# Beneficial Effects of Remote Medical Care for Patients with Hereditary Hemorrhagic Telangiectasia during the COVID-19 Pandemic

**DOI:** 10.3390/jcm10112311

**Published:** 2021-05-25

**Authors:** Eleonora Gaetani, Fabiana Agostini, Luigi Di Martino, Denis Occhipinti, Giulio Cesare Passali, Mariaconsiglia Santantonio, Giuseppe Marano, Marianna Mazza, Roberto Pola

**Affiliations:** 1Multidisciplinary Gemelli Group for HHT, Fondazione Policlinico Universitario A. Gemelli IRCCS, Università Cattolica del Sacro Cuore, 00168 Rome, Italy; fabiana.agostini@libero.it (F.A.); luigidimartino7@gmail.com (L.D.M.); denis.occhipinti@gmail.com (D.O.); GiulioCesare.Passali@unicatt.it (G.C.P.); mariaconsiglia.santantonio@policlinicogemelli.it (M.S.); giuseppemaranogm@gmail.com (G.M.); marianna.mazza@policlinicogemelli.it (M.M.); roberto.pola@unicatt.it (R.P.); 2Department of Translational Medicine and Surgery, Fondazione Policlinico Universitario A. Gemelli IRCCS, Università Cattolica del Sacro Cuore, 00168 Rome, Italy; 3Division of Otorhinolaryngology, Fondazione Policlinico Universitario A. Gemelli IRCCS, Università Cattolica del Sacro Cuore, 00168 Rome, Italy; 4Institute of Psychiatry and Psychology, Fondazione Policlinico Universitario A. Gemelli IRCCS, Università Cattolica del Sacro Cuore, 00168 Rome, Italy; 5U.P. ASPIC Università Popolare del Counseling, 00145 Rome, Italy

**Keywords:** hereditary hemorrhagic telangiectasia, COVID-19, telemedicine, remote consultation, epistaxis, quality of life

## Abstract

Background: Hereditary hemorrhagic telangiectasia (HHT) needs high-quality care and multidisciplinary management. During the COVID-19 pandemic, most non-urgent clinical activities for HHT outpatients were suspended. We conducted an analytical observational cohort study to evaluate whether medical and psychological support, provided through remote consultation during the COVID-19 pandemic, could reduce the complications of HHT. Methods: A structured regimen of remote consultations, conducted by either video-calls, telephone calls, or e-mails, was provided by a multidisciplinary group of physicians to a set of patients of our HHT center. The outcomes considered were: number of emergency room visits/hospitalizations, need of blood transfusions, need of iron supplementation, worsening of epistaxis, and psychological status. Results: The study included 45 patients who received remote assistance for a total of eight months. During this period, 9 patients required emergency room visits, 6 needed blood transfusions, and 24 needed iron supplementation. This was not different from what was registered among the same 45 patients in the same period of the previous year. Remote care also resulted in better management of epistaxis and improved quality of life, with the mean epistaxis severity score and the Euro-Quality of Life-Visual Analogue Scale that were significantly better at the end than at the beginning of the study. Discussion: Remote medical care might be a valid support for HHT subjects during periods of suspended outpatient surveillance, like the COVID-19 pandemic.

## 1. Introduction

Hereditary hemorrhagic telangiectasia (HHT) is a dominantly inheritable rare disease, characterized by the presence of multiple arteriovenous malformations (AVMs), leading to a wide variety of clinical manifestations, such us spontaneous and recurrent epistaxis, gastrointestinal bleeding, cerebral abscess or stroke due to paradoxical embolism from pulmonary AVMs, and intracerebral hemorrhage [1,2]. As with many other rare diseases, patients with HHT often experience life-long disability, life-threatening conditions, and a severely impacted quality of life [3,4]. They require many different types of medical services, including emergency rooms visits, urgent laboratory tests, blood transfusions, specific therapies, and psychological support. More generally, they need high-quality care and careful multidisciplinary follow-up and management [5].

Since the beginning of the COVID-19 pandemic, unprecedented public health measures have been undertaken worldwide to reduce the spread of the infection. Many hospitals have reduced or closed healthcare services for outpatients, including screening examinations and follow-up visits for subjects affected by HHT. The burden has been high also at the social and psychological level, especially when associated with the confinement measures imposed by many governments [6,7]. In this scenario, it is mandatory to develop alternative methods to provide assistance to patients, in particular those with peculiar needs [8,9,10,11,12]. In the specific field of rare diseases, the development of telemedicine and remote consultation is fundamental, as many of these patients have been experiencing a feeling of fear and abandon during the COVID-19 pandemic, with many cases of poor compliance to therapy and discontinuation of important treatments [13,14].

In this study, we conducted an analytical observational cohort study to evaluate whether a regimen of medical and psychological support provided to HHT patients through remote consultation during the COVID-19 pandemic has a positive impact on the disease, with reduced need of emergency room visits and hospitalizations, better management of epistaxis, and improvement of quality of life.

## 2. Materials and Methods

We contacted all patients with a definite diagnosis of HHT by e-mail and/or telephone [15] who had been followed at our HHT center for at least one year and asked them if they were willing to participate to the study. We did not contact patients below the age of 18. The patients who agreed to participate signed an informed consent form. The study started on 1 June 2020 and ended on 31 January 2021. The study was approved by the Ethics Committee of the “Fondazione Policlinico Universitario A. Gemelli IRCCS” (Rome, Italy) (protocol number 0020292/20, ID 3194).

Participating patients received weekly or biweekly clinical evaluations, depending on individual needs, by the means of remote consultations, conducted by either video calls, telephone calls, and/or e-mails. Based on individual needs, clinical evaluations were performed by one or more doctors of the multidisciplinary HHT center, including, among others, specialists in otorhinolaryngology, internal medicine, gastroenterology, neurology, and psychiatry. During remote consultations, doctors evaluated the general conditions of the individual patient, inspected the results of laboratory and radiological exams, provided clinical advices, and, if necessary, prescribed additional tests and medications.

The impact of remote medical assistance was assessed every two weeks until the conclusion of the study by determining the number of emergency room visits and hospitalizations, the need of blood transfusions, and the need of iron supplementation (either intravenous or oral). Results were compared with the data recorded in our electronic database in the same patients in the same period of time of the previous year (1 June 2019–31 January 2020). In order to assess the impact of remote consultation on the management of epistaxis, we recorded the epistaxis severity score (ESS), an internationally recognized score for nosebleed in HHT [16]. The score was measured in each patient at the beginning and at the end of the study. We also assessed the grade of overall health self-perception at the beginning and at the end of the study, using the Euro-Quality of Life-Visual Analogue Scale (EQ-VAS) [17].

SPSS 23.0 was used to perform statistical analysis. Results were expressed as mean ± SD or *n* (%). Comparisons between groups were made by χ^2^ test. *p* values less than 0.05 were considered statistically significant.

## 3. Results

A total of 126 patients met the criteria to be included in the study and were asked to participate. Of these, 45 patients (35.7%) accepted. Their demographical and clinical characteristics are presented in Table 1.

There were 21 males and 24 females with a mean age of 56.7 ± 16.1 years. All patients (100.0%) had history of epistaxis. Forty-two patients (93.3%) had mucocutaneous telangiectases. In total, 29 patients (64.4%) had one or more visceral AVMs. There were 17 patients with pulmonary AVMs (on a total of 35 who had been screened for their presence by transthoracic contrast echocardiography or CT scan of the chest), 15 patients with hepatic AVMs (on a total of 36 who had been screened for their presence by either abdominal ultrasound, CT scan, and/or MRI), and 4 patients with cerebral vascular malformations (on a total of 34 who had been screened for their presence by either CT scan and/or MRI of the brain); thirteen patients had history of gastrointestinal bleeding. Genetic tests were available for 37 patients: pathogenic mutations of the ENG gene were present in 9 patients, while pathogenic mutations of the ACVRL1 gene were present in 27 patients. A variant of unknown significance of the ACVRL1 gene was present in one patient. The mean hemoglobin level at the beginning of the study was 11.7 ± 2.4 g/dL.

During the eight-month duration of the study, we recorded the following (Table 2): 9 patients (20.0%) needed emergency room visits and/or hospitalizations; 6 patients (13.3%) needed blood transfusions; and 24 patients (53.3%) needed iron supplementation. There were no significant differences with the same period of the previous year, in which 11 patients (24.4%) needed emergency room visits, 6 patients (13.3%) needed blood transfusions, and 15 patients (33.3%) needed iron supplementation. The only significant difference was the number of patients who received oral iron therapy, which was significantly higher during the period of observation of the study than in the previous year (18/45 vs. 4/45, respectively, *p* < 0.02). Consistently, the number of patients who required intravenous iron infusion during the period of observation of the study (6/45) was lower than the year before (11/45).

Regarding epistaxis (Table 3), we found that the mean ESS was significantly lower at the end than at the beginning of the study (3.0 ± 1.6 vs. 4.4 ± 2.4 respectively, *p* < 0.01). Thirty-two patients (71.1%) reported clinically relevant episodes of epistaxis and required medical advice for the correct management of nosebleed during the study period. The mean ESS was significantly lower at the end than at the beginning of the study also in this subgroup of patients. Indeed, after receiving remote clinical advice about the management of nosebleed through telephone calls and video calls, all these patients reported improvement of symptoms and better epistaxis self-management.

Regarding quality of life, the mean EQ-VAS was 73.1 ± 13.1 at the beginning of the study (Table 4). In the absence of an established pathological threshold for EQ-VAS in HHT patients, we decided to use a value of 70 points to distinguish between subjects with normal and pathological EQ-VAS in our population. This was based on unpublished results from our group that identified 70 points as the median value of EQ-VAS among the HHT patients followed at our multidisciplinary center. Using this threshold, we found 34 patients (75.6%) with EQ-VAS within the normal range and 11 patients (24.4%) with pathological EQ-VAS. Remote psychological support and counselling therapy were offered to the 11 patients with pathological EQ-VAS at the beginning of the study. Five of these patients accepted to adhere while 6 refused. All 5 patients who accepted to receive remote psychological support and counselling therapy reached EQ-VAS values within the normal range by the end of the study. Among the 6 patients who did not accept to receive remote psychological support, only one exhibited normal EQ-VAS value at the end of the study while 3 patients had stable EQ-VAS and 2 had worse EQ-VAS. Regardless of the initial value, 14 patients (31.1%) showed improvement in EQ-VAS at the end of the study compared to the beginning of the study, 25 patients (55.6%) remained stable, and 6 patients (13.3%) reported worsening of EQ-VAS compared to the beginning of the study. Notably, EQ-VAS was higher among the 13 patients who did not have clinically relevant episodes of epistaxis (84.6 ± 8.0 points) compared to the 32 patients with severe epistaxis (70.0 ± 15.7 points). At the end of the study, 18 of the 32 patients with severe epistaxis reported an improvement in EQ-VAS.

## 4. Discussion

This study is novel and relevant for several reasons. First, it demonstrates that medical care may be provided remotely in an efficacious manner to patients affected by HHT and therefore remote consultations and telemedicine are potential alternatives to classical in-hospital follow-up visits in these patients. Second, it shows that psychological support may be provided remotely with significant beneficial effects. In addition, it provides evidence that correct management of epistaxis can be done at home by patients themselves or their caregivers, without impact on the number of emergency room visits or hospitalization. In addition, anemia may be successfully treated at home, through remote supervision and consultation, with early detection of either hemoglobin decrement and/or iron deficiency and prompt prescription of iron supplementation. Finally, this study demonstrates that personalized regimens of remote consultations have a positive impact on quality of life in HHT patients. 

Some aspects of this study deserve additional considerations. One is that remote consultation was associated with a significant increment of the proportion of patients who received a prescription of oral iron supplementation. Indeed, 40% of patients were prescribed oral iron supplementation in 2020, while only 8.8% of them received the same prescription the year before. This was also associated with a concomitant decrease of the proportion of patients who required intravenous iron supplementation (13.3% in 2020 vs. 24.4% in 2019). A possible explanation of this finding is that remote medical consultation, which was provided frequently and periodically in our study, allowed early detection of even small decrements of iron and/or hemoglobin levels and prompt therapeutic intervention. The efficacy of such approach is demonstrated by the fact that the need of blood transfusions was similar between 2019 and 2020, with also no differences in terms of emergency room visits and hospitalizations.

Another aspect that deserves attention is the satisfactory management of epistaxis that it was possible to obtain through remote consultation. Epistaxis is the disorder with the most disrupting effect on the lives of patients with HHT, both at the physical, emotional, social, and professional level [18,19]. In these patients, remote consultation always included a specialist in otorhinolaryngology, who provided practical advice on the best strategies for nasal mucosal care to prevent epistaxis and manage nosebleeds at home, along with the prescription of specific therapies, if needed. It is remarkable that there was a significant improvement of the ESS, not only in the whole study population, but also in those subjects who had clinically relevant episodes of epistaxis at the beginning of the study.

Attention should be paid also to the psychological status of HHT patients, especially in a critical period, such as the COVID-19 pandemic. Social distancing and isolation, combined with fear of contracting COVID-19, could have a significant impact on the psychological status of patients suffering from a rare disease [20]. Moreover, some studies showed that HHT patients could suffer higher levels of psychological distress than the general population, mainly because of chronic anemia and recurrent episodes of nosebleeds [3,4,20,21]. Even in our population, patients without clinically relevant episodes of epistaxis reported significantly higher EQ-VAS values than patients with severe epistaxis. Psychological support through remote consulting and counseling led to the maintenance of the mean values of EQ-VAS of our population within the normal range.

Our study has some limitations. It has a small sample size with a relatively short observation time. However, HHT is a rare disease and the COVID-19 emergency does not allow long observation times. Another limitation is that only about 35% of the patients who were asked to participate accepted to be included in the study. It is possible that these were the most motivated patients and that patients with poor compliance did not agree to participate. This might be a selection bias. Therefore, we are not sure that our results may be applied to the whole HHT population. On the other hand, it is important to point out that many of the patients that were included in the study had complicated HHT phenotypes, including severe epistaxis, chronic anemia, history of gastrointestinal bleeding, and pathological EQ-VAS. Another limitation is that patients who agreed to participate received clinical evaluations which were, in many cases, more frequent than those that they received the year before. Therefore, it cannot be excluded that the clinical improvement that we observed might depend, at least in part, by the frequency of the clinical assistance that was provided to patients and not fully to remote care itself.

In conclusion, this is the first study focused on providing remote medical care to subjects with HHT. Our data show that remote consultation might be a valid support for physicians, caregivers, and patients during periods of suspended surveillance, like the COVID-19 pandemic.

## Figures and Tables

**Table 1 jcm-10-02311-t001:** Characteristics of the study population (*n* = 45).

Mean age (years ± SD)	56.7 ± 16.1
Gender (male/female ratio)	21/24
ENG pathogenetic mutations (*n*/screened)	9/37
ACVRL1 pathogenetic mutations (*n*/screened)	27/37
Epistaxis (*n*/total)	45/45
Mucocutaneous telangiectases (*n*/total)	42/45
Family history of HHT (*n*/total)	45/45
Pulmonary AVMs (*n*/screened)	17/35
Hepatic AVMs (*n*/screened)	15/36
Cerebral AVMs (*n*/screened)	4/34
Previous gastrointestinal bleeding (*n*/total)	13/45
Anemia at the beginning of the study (*n*/total)	21/45
Mean Hb levels (g/dL) at the beginning of the study (mean ± SD)	11.7 ± 2.4

**Table 2 jcm-10-02311-t002:** Outcomes before and during the COVID-19 pandemic.

	June 2019– January 2020	June 2020– January 2021	*p*
Patients who required emergency room visits or hospitalization, *n* (%)	11 (24.4)	9 (20.0)	ns
Patients who required blood transfusions, *n* (%)	6 (13.3)	6 (13.3)	ns
Patients who required iron supplementation, *n* (%) - Oral iron supplementation, *n* (%) - Intravenous iron supplementation, *n* (%)	15 (33.3)	24 (53.3)	ns
	4 (8.8)	18 (40.0)	<0.02
	11 (24.4)	6 (13.3)	ns

**Table 3 jcm-10-02311-t003:** Epistaxis severity score (ESS) during the study.

Patients with clinically relevant episodes of epistaxis (ESS ≥ 2) (*n*/total)	32/45
Mean ESS at beginning vs. at the end of the study: - in the study population (mean ± SD) - in subjects with clinically relevant episodes of epistaxis (mean ± SD)	
	4.4 ± 2.4 vs. 3.0 ± 1.6 (*p* < 0.01)
	5.5 ± 1.9 vs. 3.6 ± 1.4 (*p* < 0.01)

**Table 4 jcm-10-02311-t004:** EQ-VAS values during the study.

Patients with normal EQ-VAS at beginning vs. end of study (*n*/total)	34/45 vs. 40/45
Patients with pathological EQ-VAS at beginning vs. end of study (*n*/total)	11/45 vs. 5/45
Mean of EQ-VAS: - in the study population (mean ± SD) - in patients with clinically relevant epistaxis (mean ± SD) - in patients without clinically relevant epistaxis (mean ± SD)	
	73.1 ± 13.1
	70.0 ± 15.7
	84.6 ± 8.0
- EQ-VAS improved at the end of the study (*n*/total) - EQ-VAS stable at the end of the study (*n*/total) - EQ-VAS worsened at the end of the study (*n*/total)	14/45
	25/45
	6/45

## Data Availability

The data presented in this study are available on request from the corresponding author. The data are not publicly available due to ethical and privacy restrictions.

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
