# Peer review of "Beneficial Effects of Remote Medical Care for Patients with Hereditary Hemorrhagic Telangiectasia during the COVID-19 Pandemic"

_jcm, 2021, doi:10.3390/jcm10112311_

Round 1

Reviewer 1 Report

The authors describe the efficiency of remote care in times of the Covid pandemic. Although the numbers are low and the outcomes may be not very exciting, the message of the paper is still interesting for colleagues taking care of HHT patients.

The paper is well written, but too many "also" in the first paragraph of the discussion. That should be adjusted.

And 1 question should be addressed: could the decrease of epistaxis also be related to seasonal influence?

Author Response

Answers to Reviewer 1

“The paper is well written, but too many "also" in the first paragraph of the discussion. That should be adjusted.”

The first paragraph of the discussion has been edited as requested by the Reviewer (page 5, lines 171-182).

“One question should be addressed: could the decrease of epistaxis also be related to seasonal influence?”

It is true that there might be a seasonal variation in epistaxis. Indeed, it is known that epistaxis occurs more commonly during the winter and that the cold season is associated with increased epistaxis frequency and severity. However, our study started in June and ended in January, and we registered a lower Epistaxis Severity Score (ESS) at the end of the study (thus in the cold season) compared to the beginning of the study (which was during the summer). Therefore, it is unlikely that there is a seasonal influence on the results of this study regarding epistaxis.

Reviewer 2 Report

Dear authors,

I read with great interest your manuscript entitled “Beneficial Effects of Remote Medical Care for Patients with HHT during COVID-19 Pandemic” and I share with you the responsibility of assuring physical and psychological welfare to our patients, even in adverse conditions as the pandemic we are suffering since last year. On this matter, new technologies and telemedicine have experienced a boost that will probably last beyond the pandemic and therefore, reporting positive experiences is good to reinforce the idea that remote medical care can be an alternative to usual outpatient care. However, as a clinician, I have big concerns about the results of your study as it may drive to misleading conclusions.

To start with, I miss the total number of patients you usually follow up in your unit to know the proportion of them that accepted participating the study. As an observational study, you should follow STROBE recommendations.

After, I think is important for the reader to know which is the usual frequency you see these patients in the outpatient setting as you inform in page 2, line 74 that “participating patients received weekly or biweekly clinical evaluations”. Isn’t it possible that they got contacted more often than they used to before the pandemic? And, if this is the case, could this be the reason behind the clinical and psychological improving? In addition, if follow ups are not strictly the same in terms of frequency, how can the results be attributable to remote medical care?

Moreover, one of the results you claim is the improvement of iron oral therapy reflected in the reduction of intravenous iron infusions. When the reviewer reads this result, tends to think that these patients were previously under-treated and were not receiving optimal oral iron therapy before switching to intravenous iron therapy, as recommended in the international HHT treatment guidelines.

Finally, I am not quite familiarized with the EQ-VAS, but in the cite you provide in the text (Feng Y, Parkin D, Devlin NJ. Assessing the performance of the EQ-VAS in the NHS PROMs Programme. Qual Life Res 289 2014 Apr, 23(3), 977-89) I could not find any mention to a burden of 70 points to classify as pathological or not. You should clarify why this number was the chosen as a threshold.

Because of all these reasons and although I believe that reporting positive experiences in remote medical care is useful to encourage telemedicine, I strongly recommend to redesign the whole manuscript, as I feel that the improvement of clinical variables you report and the conclusions derived from them are not prudent as they come from deficient methodology.

Author Response

“I miss the total number of patients you usually follow up in your unit to know the proportion of them that accepted participating the study. As an observational study, you should follow STROBE recommendations.”

In the revised version of the manuscript, we clearly specify the number of patients who were contacted and asked to participate to the study (page 3, lines 102-103). The study follows the STROBE recommendations for observational studies. The observational nature of the study is clearly stated in the revised version of the abstract and the manuscript (line 22 and line 62).

“ I think is important for the reader to know which is the usual frequency you see these patients in the outpatient setting as you inform in page 2, line 74 that “participating patients received weekly or biweekly clinical evaluations”. Isn’t it possible that they got contacted more often than they used to before the pandemic? And, if this is the case, could this be the reason behind the clinical and psychological improving? In addition, if follow ups are not strictly the same in terms of frequency, how can the results be attributable to remote medical care?”

The Reviewer raises a good point. This is a weakness of the study that we acknowledge in the revised version of the manuscript (page 6, lines 228-232).

“One of the results you claim is the improvement of iron oral therapy reflected in the reduction of intravenous iron infusions. When the reviewer reads this result, tends to think that these patients were previously under-treated and were not receiving optimal oral iron therapy before switching to intravenous iron therapy, as recommended in the international HHT treatment guidelines.”

We have edited the manuscript to clarify that remote medical consultation allowed early detection of even small decrements of iron and/or haemoglobin levels and prompt therapeutic intervention, which was mainly based on oral iron supplementation, with consensual reduction of the use of intravenous iron therapy (page 5, lines 184-195).

"I am not quite familiarized with the EQ-VAS, but in the cite you provide in the text (Feng Y, Parkin D, Devlin NJ. Assessing the performance of the EQ-VAS in the NHS PROMs Programme. Qual Life Res 289 2014 Apr, 23(3), 977-89) I could not find any mention to a burden of 70 points to classify as pathological or not. You should clarify why this number was the chosen as a threshold.”

There is not an established pathological threshold for EQ-VAS in HHT patients. We decided to use a value of 70 points based on our personal experience with the HHT population and unpublished results from our group that have identified 70 points as the median value of EQ-VAS among the HHT patients followed at our multidisciplinary center. The reason of our choice is explained in the revised version of the manuscript (page 4, lines 147-151).

Round 2

Reviewer 2 Report

All comments have been properly ammended.

Author Response

We are pleased that the Reviewer is satisfied with our answers to her/his comments and are grateful for her/his positive feedback.